# The Impact on the Clinical Prognosis of Low Serum Selenium Level in Patients with Severe Trauma: Systematic Review and Meta-Analysis

**DOI:** 10.3390/nu14061295

**Published:** 2022-03-18

**Authors:** Shang-Yu Chan, Chih-Po Hsu, Chun-Hsiang Ou Yang, Chia-Cheng Wang, Yu-Tung Wu, Chih-Yuan Fu, Chi-Hsun Hsieh, Chi-Tung Cheng, Wei-Cheng Lin, Jen-Fu Huang, Chien-Hung Liao

**Affiliations:** 1Division of Trauma and Emergency Surgery, Chang Gung Memorial Hospital, Chang Gung University, Taoyuan 33328, Taiwan; shengyu.chan@gmail.com (S.-Y.C.); chihpo1227@gmail.com (C.-P.H.); detv090@gmail.com (C.-H.O.Y.); m0827@cgmh.org.tw (C.-C.W.); overwinterwu@gmail.com (Y.-T.W.); drfu5564@gmail.com (C.-Y.F.); hsieh0818@cgmh.org.tw (C.-H.H.); atong89130@gmail.com (C.-T.C.); surgymet@gmail.com (C.-H.L.); 2Department of Electrical Engineering, Chang Gung University, Taoyuan 33328, Taiwan; weiclin@mail.cgu.edu.tw

**Keywords:** selenium, trauma, injury, mortality

## Abstract

This study was designed to examine the most up-to-date evidence about how low plasma selenium (Se) concentration affects clinical outcomes, such as mortality, infectious complications, and length of ICU or hospital stay, in patients with major trauma. We searched three databases (MEDLINE, EMBASE, and Web of Science) with the following keywords: “injury”, “trauma”, “selenium”, and “trace element”. Only records written in English published between 1990 and 2021 were included for analysis. Four studies were eligible for meta-analyses. The results of the meta-analysis showed that a low serum selenium level did not exert a negative effect on the mortality rate (OR 1.07, 95% CI: 0.32, 3.61, *p* = 0.91, heterogeneity, I^2^ = 44%). Regarding the incidence of infectious complications, there was no statistically significant deficit after analyses of the four studies (OR 1.61, 95% CI: 0.64, 4.07, *p* = 0.31, heterogeneity, I^2^ = 70%). There were no differences in the days spent in the ICU (difference in means (MD) 1.53, 95% CI: −2.15, 5.22, *p* = 0.41, heterogeneity, I^2^ = 67%) or the hospital length of stay (MD 6.49, 95% CI: −4.05, 17.02, *p* = 0.23, heterogeneity, I^2^ = 58%) in patients with low serum Se concentration. A low serum selenium level after trauma is not uncommon. However, it does not negatively affect mortality and infection rate. It also does not increase the overall length of ICU and hospital stays.

## 1. Introduction

Selenium (Se) is an important element in the regulation of immune and inflammation pathways in human body. It is also an essential micronutrient that is required for 25 proteins in the human body [1,2,3]. These selenoproteins play critical physiological roles in immune function, endocrine function, wound healing, and protein synthesis [1,2,3], and elicit antioxidative effects and thyroid metabolism [4,5,6]. Moreover, studies have shown that Se is beneficial in preventing infection and enhancing the recovery of autoimmune thyroiditis, malignancy, and cardiac disease [7,8]. Se deficiency is associated with an inadequate health status, especially pregnant women, women during lactation time, and geriatric populations [9].

Trauma accounts for one of the leading causes of global death [10]. The initial mortality is associated with the trauma severity [11]. However, deaths that occur between days and weeks after injury are often related to the inflammatory process after trauma [12]. During this period of time, multiorgan failure developing from sepsis and inflammation becomes the major cause of mortality [13,14]. Cascading inflammatory responses and severe metabolic malfunction have been encountered after major trauma [15]; moreover, the cascade of the inflammatory response is further aggravated by infection or sepsis [15]. A rapid reduction in serum Se level has been reported after trauma events [16]. This phenomenon of declined Se level might be a hemodilution event after hemorrhage and massive fluid resuscitation [17,18,19]. Se deficiency has been known to decrease the function of immune cells during activation, differentiation, and proliferation and could negatively affect the human body in sepsis, wound healing, and muscle catabolism [8]. Furthermore, low serum Se levels are often reported in critically ill patients and patients with severe trauma, which are related to more oxidative stress and inflammatory molecules, and are associated with a worse prognosis [5,20,21]. Patients who sustain major trauma with lower serum Se levels might have more infection-related adverse events and thus lead to worse outcomes; however, the relationship between low plasma Se levels in patients with major trauma and sepsis remains unclear. To the best of our knowledge, there are no critical and thorough analyses to demonstrate the relationship between clinical outcomes and low plasma Se levels in patients with severe trauma.

The goal of this review is to assess current evidence regarding the effect of low serum Se concentration on mortality, length of intensive care unit (ICU) stay, hospital stay, and infection rates in patients with severe trauma.

## 2. Materials and Methods

### 2.1. Protocol and Registration

The literature search was conducted based on the 2019 Preferred Reporting Items for Systematic Reviews and Meta-Analyses (PRISMA) statement and it was also registered in Prospero (registered no. 297041).

### 2.2. Eligibility Criteria

#### 2.2.1. Population

This review enrolled studies that included patients with severe trauma admitted to the hospital and which reported their serum Se level.

#### 2.2.2. Intervention and Comparison

Enrolled patients were classified by their level of serum Se. Comparisons between patients with low or normal serum Se concentrations were made.

#### 2.2.3. Outcome Measures

The primary outcomes of this review were focused on mortality and infectious complications.

#### 2.2.4. Studies

Randomized controlled trials (RCTs), experimental and epidemiological study designs (including non-randomized controlled trials), and prospective and retrospective cohort studies were considered for eligibility for this review.

### 2.3. Information Sources and Search

We searched three online databases (MEDLINE, EMBASE, and Web of Science) for eligible studies. Records written in English published between January 1990 and December 2020 were considered. The record searches included the following keywords: “trace element”, “selenium”, “injury”, and “trauma”. The details of the research are listed in Table 1. Furthermore, the references of relevant articles were evaluated for eligibility.

### 2.4. Study Selection

Two independent reviewers (Huang, J.F. and Chan, S.Y.) screened the titles and abstracts to determine whether the studies were eligible. Studies reporting clinical outcomes along with serum Se concentration were enrolled since this allowed for calculating the relative risk of the studied population.

### 2.5. Data Collection Process and Quality Assessment

Two independent reviewers (Huang, J.F. and Chan, S.Y.) extracted the data from selected studies. The data included serum Se concentration, demographic data (age, gender), study designs, and clinical outcomes (mortality, infectious complications, morbidity, ICU length of stay (LOS), hospital LOS). When targeted data were missing, we tried to calculate them from data available from the records.

The authorship or institution was not blinded to the reviewers. Disagreements during the whole review were referred to a third reviewer (Hsu, C.P.). The Newcastle Ottawa Scale was used to determine the quality of the enrolled studies [22].

### 2.6. Data Synthesis

All relevant data were pooled for the meta-analysis with Revman 5 software, Biostat, Englewood, NJ, USA. Effect sizes were reported as odds ratios (OR) with 95% confidence intervals (CI) for mortality and infectious complications. The difference in means (MD) and 95% CI for LOS was calculated for numeric parameters. For assessment of heterogeneity and inconsistency, we used the standard χ^2^ test and the I^2^ test. Since only a small number of studies were included, publication bias was not calculated due to the insufficient statistical power to detect the asymmetry [23]. *p <* 0.05 was defined as statistically significant.

## 3. Results

### 3.1. Study Selection

After searching three databases using the keywords, 1064 records were identified. Twenty full-text articles were assessed for eligibility after screening of abstract, sixteen of which were excluded according to the following reasons. Nine articles lacked a relevant outcome, six articles included an inappropriate population, and one article was written with a duplicated cohort (Figure 1). The remaining four studies were considered eligible for the methodological quality requirement and were enrolled in the meta-analysis [4,21,24,25]. The study selection process is illustrated in Figure 1.

### 3.2. Study Characteristics

The included studies were three prospective, randomized, blinded, control trials and one non-randomized experimental trial. The study designs and data characteristics of the selected studies are shown in Table 2. A total of 342 participants were enrolled from the studies in this review.

### 3.3. Analysis

Four studies reported the effect of low serum Se concentration on the mortality rate of patients who had sustained major trauma. The pooled results showed that low serum Se concentration did not affect the mortality (Figure 2, OR 1.07, 95% CI: 0.32, 3.61, *p* = 0.91, heterogeneity, I^2^ = 44%). Regarding infectious complications, there was no statistically significant difference between two groups, but it was found to slightly favor those with normal serum Se concentrations (Figure 3, OR 1.61, 95% CI: 0.64, 4.07, *p* = 0.31, heterogeneity, I^2^ = 70%). When we focused on pneumonia, no statistically significant differences between patients with low or normal serum Se concentrations were found (Figure 4, OR 1.77, 95% CI: 0.75, 4.20, *p* = 0.19, heterogeneity I^2^ = 65%).

Four studies reported time spent in intensive care unit (ICU) and hospital. Both the days spent in the ICU and the hospital were not statistically significant (Figure 5 and Figure 6, *p* = 0.41 and 0.23, respectively). Three studies report days of ventilator use. There was also no statistically significant difference between patients with low or normal serum Se concentrations and days of ventilator use (Figure 7, MD 0.98, 95% CI −1.27, 3.23, *p* = 0.39).

## 4. Discussion

In this meta-analysis, the clinical outcomes of low serum Se concentration on patients with severe trauma were examined. The current evidence demonstrates that a low serum Se concentration does not increase the mortality rate. It also does not have a negative effect on the length of ICU and hospital stays for patients with severe trauma. Although the mortality and infection rates were higher in patients with low serum Se concentrations, no statistically significant differences were found. The anti-inflammatory effect of Se might play a role in patients with severe trauma, thus reducing the mortality and infectious complications during the recovery phase [16,19,20]. Furthermore, the Se supplementation was associated with regulation of thyroid hormone and glutathione peroxidase concentrations, which help patients to deal with hypermetabolic status. However, in this review, the protective effects do not to appear to constitute a significant difference.

Low serum Se concentrations are often found in patients with severe trauma. Some studies reported that low serum Se levels are related to more complications in patients with severe burns or multiple traumas [21,26]. In several studies, it was found that Se supplementation can also decrease mortality and reduce the infection rate in the event of major trauma, burns, and critical illness [18,27,28]. From this perspective, a deficiency of Se might impact the immune status in this type of patient and sequentially influence their final prognosis. However, in this review, a low serum selenium level had an insignificant effect on length of hospital or ICU stay, infection rate, and mortality rate. This may indicate that the measurements of the serum level of Se might not be accurate enough to assist clinical judgment. Se supplementation, whether administered in the presence of an Se deficiency or not, can help trauma patients’ prognoses, based on the findings of the previous literature [27,29], if patients have undergone trauma and are experiencing a sequential post-traumatic immune reaction, beginning with a state of hyper-inflammation, followed by a compensatory anti-inflammatory immune response [15,16]. In this situation, patients might need more Se and associated trace elements than usual, as even patients with normal serum Se levels suffer from insufficient Se catabolism in this situation. However, it is important to note that more prospective controlled studies are required to support this finding in the future.

Few studies have reported the relationship between the clinical outcomes and low serum Se concentrations in patients with severe trauma and there are no large-scale clinical trials available. Importantly, our review contributes to the existing literature by providing more evidence to this field. However, there were limitations to this review. First, only a few studies with small sample sizes were included, which could lead to type II errors. Second, the included studies were written between 2001 to 2019; improvement in the treatment of trauma over time could be another confounding factor. Third, routine Se examination is not frequent in current practice, leading to limitations in the study evaluation. Fourth, the low plasma Se concentration found in these studies may only be related to inflammation rather than true Se deficiency. In the future, large-scale multicenter studies that consider severity of injury, Se biochemical measurement, and the related physiological effects are essential to provide definitive evidence. Fifth, there were numerous studies that discussed the molecular and biologic effects of the serum Se concentration and selenoprotein; however, in this review, we focused on clinical studies. As a result, we could not include those studies in order to expand our analysis.

## 5. Conclusions

This review indicates that a low plasma Se concentration after trauma is not uncommon. However, it does not negatively affect mortality and infection rates. It also does not increase the overall length of ICU and hospital stay.

## Figures and Tables

**Figure 1 nutrients-14-01295-f001:**
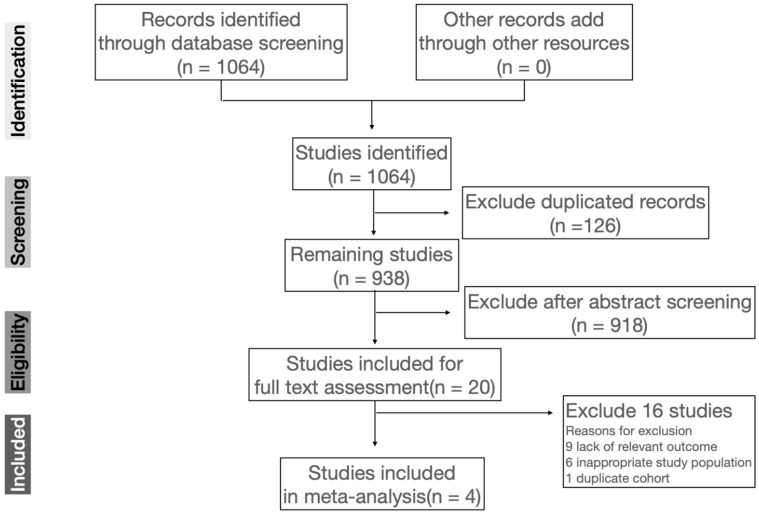
The protocol of this systematic review.

**Figure 2 nutrients-14-01295-f002:**
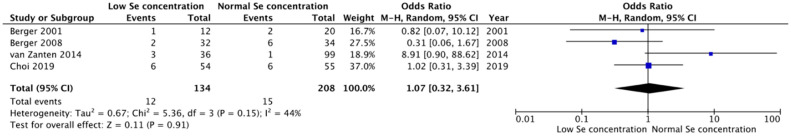
Forest plot of the impact of low serum Se concentration on mortality of patients with severe trauma.

**Figure 3 nutrients-14-01295-f003:**
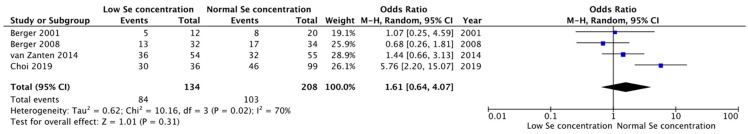
Forest plot of the impact of low serum Se concentration on the presence of infection of in patients with severe trauma.

**Figure 4 nutrients-14-01295-f004:**
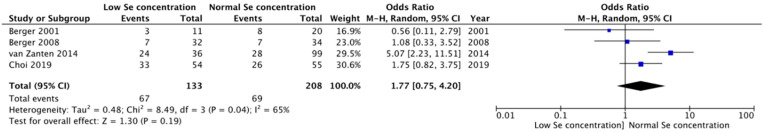
Forest plot of the impact of low serum Se concentration on the presence of pneumonia of in patients with severe trauma.

**Figure 5 nutrients-14-01295-f005:**
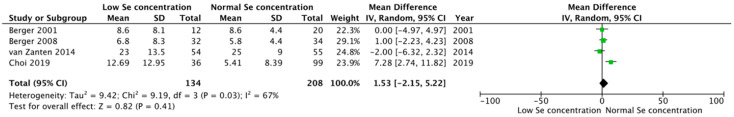
Forest plot of the impact of low serum Se concentration on the length of intensive care unit stay in patients with severe trauma.

**Figure 6 nutrients-14-01295-f006:**
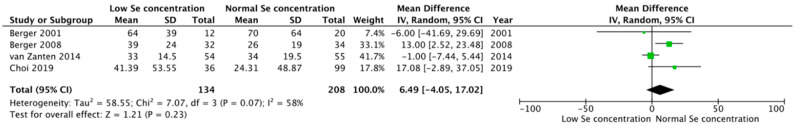
Forest plot of the impact of low serum Se concentration on the length of hospital stay in patients with severe trauma.

**Figure 7 nutrients-14-01295-f007:**
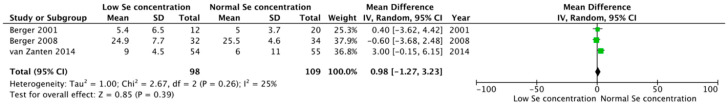
Forest plot of the impact of low serum Se concentration on the length of ventilator usage in patients with severe trauma.

**Table 1 nutrients-14-01295-t001:** Detailed database search strategy including key words and limits.

Database	Search Terms	Filters/Limits
**Pubmed**	(trace element OR selenium) AND (trauma OR injury)	Clinical Trial, Randomized Controlled Trial, Humans, English, from 1990–2021
**Embase**	(trace: ti, ab AND element: ti, ab OR selenium: ti, ab) AND (trauma: ti, ab OR injury: ti, ab)	[humans]/lim AND [english]/lim AND [clinical study]/lim AND [1990–2021]/py
**Web of Science**	((TS = (trace element OR selenium)) OR (TI = (trace element OR selenium))) AND ((TS = (trauma OR injury)) OR TI = (trauma OR injury)) NOT ALL = (in vitro OR rabbit OR rat OR animal OR mice OR mouse OR pig OR porcine OR sheep OR lamb) AND (DT = (“ARTICLE” OR “MEETING ABSTRACT” OR “PROCEEDINGS PAPER” OR “EDITORIAL MATERIAL” OR “EARLY ACCESS”))	From 1990–2021

**Table 2 nutrients-14-01295-t002:** Characteristics of included studies.

Study	Methods	Participants, Setting	Intervention	Outcome Measures	Newcastle Ottawa Scale
Berger et al. (2001) [4]	Study design: prospective RCT, DB Duration of follow-up: 20 days	Participants: Total *n* = 32 Intervention *n* = 20; placebo *n* = 12 23 males and 9 females Mean age: 42.43 ± 16.55 years Setting: SICU, Centre Hospitalier Universitaire Vaudois, Lausanne, Switzerland Inclusion criteria: Age 18–70; multiple injuries with ISS > 15; admission within first 24 h of injury	Intervention: 500 μg Se alone per day, or Se and 150 mg alpha-tocopherol, 2.6 mg Cu and 13 mg Zn per day Control: placebo	Mortality Incidence of complications and organ failure Hospital stays	8/9
Berger et al. (2008) [24]	Study design: prospective RCT, DB Duration of follow-up: 3 months after discharge	Participants: Total *n* = 66 Intervention *n* = 34; placebo *n* = 32 52 males and 14 females Mean age: 40 ± 19 years Setting: SICU, Centre Hospitalier Universitaire Vaudois, Lausanne, Switzerland Inclusion criteria: ISS > 9	Intervention: selenium 270 μg IV, zinc 30 mg IV, vitamin C 1.1 g IV, vitamin B1 100 mg IV, vitamin E 6.4 mg IV and 300 mg PO with a double-loading dose on days 1 and 2, total for 5 days plus ICU standard vitamin profile as control group Control: ICU standard vitamin profile: 500 mg vitamin C/day for 5 days and 100 mg vitamin B1/day for 3 days	Mortality Hospital stays Kidney function Subsequent organ failure Infections and pneumonia	8/9
van Zanten et al. (2014) [25]	Study design: prospective RCT, DB Duration of follow-up: Six months after start study product	Participants: Total *n* = 109 Intervention *n* = 55; control *n* = 54 87 males and 22 females Mean age: 43 years Setting: ICU, multi-country, multi-center Inclusion criteria: age ≥ 18 years, mechanically ventilated ICU patients	Intervention: Tube feed formula enriched in glutamine, vitamin C and E, selenium, zinc and EPA and DHA, and low in carbohydrate content Control product: Isocaloric standard tube feed with the same amount of protein.	Incidence of nosocomial infections and organ failure Duration of ventilation, ICU and hospital stay Mortality	9/9
Choi et al. (2019) [21]	Study design: retrospective case control study Duration of follow-up: Until discharge	Participants: Total *n* = 135 Intervention *n* = 36; control *n* = 99 99 males and 36 females Mean age: 48.5 ± 18.8 years Setting: Trauma center, Yonsei University College of Medicine, Seoul, Korea Inclusion criteria: serum selenium levels were measured within 2 days of admission	Classification by Se level Case: >70 ng/mL Control: <70 ng/mL	Infection complications, Duration of ventilation, ICU and hospital stay Mortality	4/9

## Data Availability

The data presented in this study are openly available in References [4,23,24,25].

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
