# Peer review of "The Impact on the Clinical Prognosis of Low Serum Selenium Level in Patients with Severe Trauma: Systematic Review and Meta-Analysis"

_nutrients, 2022, doi:10.3390/nu14061295_

Round 1

Reviewer 1 Report

This systematic review is a useful addition to the literature, in particular it provides evidence that can be used in the treatment of trauma patients. There are a number of points that need to be addressed:

  1. Lines 23-26 need rewording. If there is no significant difference then a statement about an effect should not be made.
  2. Line 26 and throughout the m/s. There is no proof that patients with trauma are Se deficient per se. The fall in plasma Se concentration may be similar to falls in Fe and Zn seen with inflammation. Therefore it would be better to refer to low plasma Se concentration rather than inferring that the patients are Se deficient.
  3. Line 32. Needs rewording.
  4. Line 33. I cannot access Ref 1 but it is generally accepted that there are 25 genes that encode selenoproteins (see Santesmasses & Gladyshev Int J Mol Sci 2021;22:11593) not >25. 
  5. Line 68. Needs rewording.
  6. Lines 70-71. Unclear.
  7. Lines 107-122. Delete text after "significant" as these are instructions for authors.
  8. :Line 126. Define "rationally".
  9. Lines 134-137. Delete instructions for authors.
  10. Lines 145 onwards. Results are overinterpreted ("slightly favour"). High heterogeneity. Cannot interpret combined effect a meaningful, so have to conclude there is no effect.
  11. Figs 2-7 titles. Change "chart" to plot.
  12. Line 168. As in point 2 above. 
  13. Line 169. What does "unexpected" mean? "U" should be lower case.
  14. Line 172. Change to patients with low plasma Se concentration.
  15. Line 180. Reword.
  16. Lines 185 onwards. This paragraph needs careful editing as it is too speculative in places.
  17. Line 199. What is "regular"?
  18. Line 204. Reword.
  19. Line 212. Low plasma Se concentration, not Se deficiency.

Author Response

  1. Lines 23-26 need rewording. If there is no significant difference then a statement about an effect should not be made.

-     Reply: Thank you for your instructive comments, we have changed the statement as your advice. Thank you again.

  1. Line 26 and throughout the m/s. There is no proof that patients with trauma are Se deficient per se. The fall in plasma Se concentration may be similar to falls in Fe and Zn seen with inflammation. Therefore it would be better to refer to low plasma Se concentration rather than inferring that the patients are Se deficient.
    • Thank you for the valuable comments. We agreed that a low serum selenium level did not equal selenium deficiency. We have changed the term as suggested. We also added an explanation in the section on the limitations of this study.
    • We also changed the group name in all plots.
  2. Line 32. Needs rewording.
    • Reply : Thank you for your comment, we reworded this sentence to be more precise as the reference. Thank you again for making our manuscript clearer to read.
  3. Line 33. I cannot access Ref 1 but it is generally accepted that there are 25 genes that encode selenoproteins (see Santesmasses & Gladyshev Int J Mol Sci 2021;22:11593) not >25.
    • Thank you for the comments. We have changed the manuscript according to the reference. Thank you again for making our manuscript clearer to read.
  4. Line 68. Needs rewording.
    • Reply : We revised this phrase to ‘patients with severe trauma;’ thank you again.
    • Line 68-69 : This review considered studies which included patients with severe trauma admitted to the hospital and reported serum Se levels.
  5. Lines 70-71. Unclear.
    • Reply : We revised this sentence to clarify this statement. Thank you again for making our article more valuable to read.
    • Line 71-72 : Enrolled patients are classified by the level of serum Se. Comparisions between patients with low or normal serum Se concentration are made.
    •  
  6. Lines 107-122. Delete text after "significant" as these are instructions for authors.
    • Thank you for the comments. We have deleted the unnecessary text.
  7. :Line 126. Define "rationally”.
    • Thank you for the comments. We have defined the rationale for selection.
    • Through database searching, we identified 1064 citations. Twenty full-text articles were assessed for eligibility, sixteen of which were rationally excluded. Nine articles lacked a relevant outcome, six articles included an inappropriate population, and one article was written with a duplicated cohort (Figure 1).
  8. Lines 134-137. Delete instructions for authors.
    • We have deleted the instructions for authors.
    •  
  9. Lines 145 onwards. Results are overinterpreted (“slightly favour”). High heterogeneity. Cannot interpret combined effect a meaningful, so have to conclude there is no effect.
    • Thank you for the comments. We have changed the statement to reflect that there is no effect. Thank you again.
    • When we focus on pneumonia, there was also no statistical difference between patients with or without Se deficiency. (Figure 4, OR1.77, 95% CI: 0.75, 4.20, p = 0.19, hetero-geneity I2 = 65%)
  10. Figs 2-7 titles. Change “chart” to plot.

-        Reply : Thank you for your comment; we have revised these words.

Line 168. As in point 2 above. 

  • Reply: Thank you for your thoughtful opinion. We agree with you about the statement of selenium deficiency; we have changed the state to “lower serum Se concentration”.
  1. Line 169. What does “unexpected” mean? “U” should be lower case.
    • We have deleted the unnecessary word “unexpected”. Thank you for the careful reviewing.
  2. Line 172. Change to patients with low plasma Se concentration.
    • We have changed the term as suggested.
  3. Line 180. Reword.
    • We have finished the editing. Thank you.
  4. Lines 185 onwards. This paragraph needs careful editing as it is too speculative in places.
    • Thank you for the insightful comments. This paragraph had been edited to avoid speculative hypothesis. We assumed that there was unmet demand for Se for the patients with major trauma. Thus, a normal serum level did not represent a sufficiency of Se.
  5. Line 199. What is "regular"?
    • We have changed to “normal”. Thank you.
  6. Line 204. Reword.
    • We have finished the editing. Thank you.
  7. Line 212. Low plasma Se concentration, not Se deficiency.

Thank you for the comments. We have changed the statement as suggested.

Reviewer 2 Report

Review nutrients- 1614108

In this review the authors aim to clarify the Impact on the Clinical Prognosis of Selenium Deficiency in Severe Trauma Patients. For this purpose they performed a meta-analysis to proof the potential efficacy of Se supplementation in severe trauma patients.

I find the manuscript interesting for the community, but there are some matters that should be discussed before publication.

First of all, the manuscript would benefit greatly from a proper grammar check.

  1. Please rephrase the way you address your patients: “patients with severe trauma” instead of “severe trauma patients”.
  2. Line 34: Why only SELENOP? There are as the author states: 25 selenoproteins? It is not only SELENOP that has effects on the mechanisms spelled out.
  3. Line 107-122: this is standard text that is in the template provided by MPDI/Nutrients. Please remove/check for completeness.
  4. Line 134-137: again text from the template!
  5. I really miss some in depth reviewing of the potential molecular mechanisms explaining the effect of Selenium (via more!! then 1 selenoprotein (SELENOP)). SELENOP is the main carrier of selenium in the circulatory system and is therefore often marked as a good surrogate for total blood selenium concentrations. It are possible a number of the other 24 that have a pronounced effect on inflammation and oxidative stress…
  6. Line 192: “the SELENOP-specific pathways” could the author please explain what he means by this?
  7. All in all this manuscript is only a very low-numbered meta-analysis and not a review at all.
  8. The small numbers found in literature contradict the conclusion. Finding only 300+ cases doesn’t reflect “Se deficiency after trauma is not uncommon”.

Author Response

In this review the authors aim to clarify the Impact on the Clinical Prognosis of Selenium Deficiency in Severe Trauma Patients. For this purpose they performed a meta-analysis to proof the potential efficacy of Se supplementation in severe trauma patients.

I find the manuscript interesting for the community, but there are some matters that should be discussed before publication.

First of all, the manuscript would benefit greatly from a proper grammar check.

Response: Thank you for the comment. We had sent the manuscript for English editing by MDPI.

  1. Please rephrase the way you address your patients: “patients with severe trauma” instead of “severe trauma patients”.
    • Thank you for the comment. We have changed to “patients with severe trauma” as suggested.
  2. Line 34: Why only SELENOP? There are as the author states: 25 selenoproteins? It is not only SELENOP that has effects on the mechanisms spelled out.
    • Thank you for the comments. We have changed to “selenoproteins” as suggested.
  3. Line 107-122: this is standard text that is in the template provided by MPDI/Nutrients. Please remove/check for completeness.
    • We have deleted unnecessary text. Thank you.
  4. Line 134-137: again text from the template!
    • We have deleted unnecessary text. Thank you.
  5. I really miss some in depth reviewing of the potential molecular mechanisms explaining the effect of Selenium (via more!! then 1 selenoprotein (SELENOP)). SELENOP is the main carrier of selenium in the circulatory system and is therefore often marked as a good surrogate for total blood selenium concentrations. It are possible a number of the other 24 that have a pronounced effect on inflammation and oxidative stress.
    • Thank you for the insightful comments. This paragraph had been edited to avoid speculative hypotheses. We assumed that there was unmet demand in Se for the patents with major trauma. Thus, a normal serum level did not represent a sufficiency of Se.
  6. Line 192: “the SELENOP-specific pathways” could the author please explain what he means by this?
    • Thank you for the insightful comments. This paragraph had been edited to avoid speculative hypotheses. We assumed that there was unmet demand in Se for the patents with major trauma. Thus, a normal serum level did not represent a sufficiency of Se.
  7. All in all this manuscript is only a very low-numbered meta-analysis and not a review at all.
    • Thank you for the comments. There was limited evidence regarding the effect of a low serum selenium level on patients with major trauma currently. A detailed review would be difficult. We provided a meta-analysis of the available studies to help the clinical practitioner understand the role of serum selenium level on prognosis.
  8. The small numbers found in literature contradict the conclusion. Finding only 300+ cases doesn’t reflect “Se deficiency after trauma is not uncommon”.
    • Thank you for the valuable comment. The true incidence of a low serum selenium level in patients sustaining major trauma remained unclear. In one of our references, Se deficiency could be found in 26% of patients with severe trauma. (Choi, S.B.; Jung, Y.T.; Lee, J.G. Association of Initial Low Serum Selenium Level with Infectious Complications and 30-Day Mortality in Multiple Trauma Patients. Nutrients 2019, 11, doi:10.3390/nu11081844) We have deleted the statement in the conclusion.
  9. We have changed “Se deficiency” to “low serum Se concentration” in our article and changed the group name in all cases to ‘plot.’

Round 2

Reviewer 2 Report

No further comments